# Corticosteroids in ARDS

**DOI:** 10.3390/jcm12093340

**Published:** 2023-05-08

**Authors:** Emmanuelle Kuperminc, Nicholas Heming, Miguel Carlos, Djillali Annane

**Affiliations:** 1Department of Intensive Care, Hôpital Raymond Poincaré, APHP University Versailles Saint Quentin-University Paris Saclay, 92380 Garches, France; 2Laboratory of Infection & Inflammation—U1173, School of Medicine Simone Veil, University Versailles Saint Quentin-University Paris Saclay, INSERM, 92380 Garches, France; 3FHU SEPSIS (Saclay and Paris Seine Nord Endeavour to PerSonalize Interventions for Sepsis), 92380 Garches, France

**Keywords:** glucocorticoids, acute respiratory distress syndrome, sepsis

## Abstract

Acute respiratory distress syndrome (ARDS) is frequently associated with sepsis. ARDS and sepsis exhibit a common pathobiology, namely excessive inflammation. Corticosteroids are powerful anti-inflammatory agents that are routinely used in septic shock and in oxygen-dependent SARS-CoV-2 related acute respiratory failure. Recently, corticosteroids were found to reduce mortality in severe community-acquired pneumonia. Corticosteroids may therefore also have a role to play in the treatment of ARDS. This narrative review was undertaken following a PubMed search for English language reports published before January 2023 using the terms acute respiratory distress syndrome, sepsis and steroids. Additional reports were identified by examining the reference lists of selected articles and based on personnel knowledge of the authors of the field. High-quality research is needed to fully understand the role of corticosteroids in the treatment of ARDS and to determine the optimal timing, dosing and duration of treatment.

## 1. Introduction

### 1.1. Sepsis—Definition, Epidemiology and Pathophysiology

Sepsis is defined as a “life-threatening organ dysfunction caused by a dysregulated host response to infection” [1]. The abnormal immune response causes systemic inflammation and organ dysfunction [2]. Sepsis is a global cause of morbidity and mortality [3]. Globally, the incidence of sepsis is estimated at 190/100,000 person-years, and mortality is between 20 and 30% [4]. The most common causes of sepsis are diarrheal disease, with 9.2 million annuals cases globally, while the most common cause of sepsis-related death is lower respiratory tract infections, with 1.8 million annuals cases globally [5]. Sepsis was recently identified as a global health priority by the World Health Organization [6]. The annual cost of sepsis in the US is over $20 billion [7]. Half of sepsis survivors experience subsequent physical and psychological disabilities, significantly altering their quality of life [8].

The pathobiology of sepsis is complex, associating immune, hormonal, energetic and metabolic anomalies. The excessive inflammatory response observed in sepsis is caused by an imbalance between two distinct pathways, on the one hand the activated inflammatory pathway, (NF-kB signaling system), and on the other hand the dysregulated hypothalamic-pituitary-adrenal-axis (HPA) response [9]. The initial host response following an insult involves recognition of danger signals by innate immune cells leading to the activation of multiple and sometimes redundant signaling pathways. The initial inflammatory response is triggered by the interaction between pathogen-associated molecular patterns (PAMPs) and pattern-recognition receptors on host immune cells. These receptors include several families of immune receptor such as Toll-like receptors (TLR), C-type lectin receptors, retinoic acid inducible gene 1-like receptors, and nucleotide-binding oligomerization domain-like receptors (NOD-like receptors), localized on cell membranes, in endosomes, or in the cytoplasm [10]. Excessive inflammation may lead to tissue damage and cell death, causing the release of additional danger molecules (damage-associated molecular patterns; DAMPs), such as the S100 protein, high mobility group box 1, extracellular RNA, DNA and histones which can further contribute to inflammation through activation of additional pattern-recognition receptors [11]. Binding of a ligand to a Toll-like receptor leads to the activation of signaling pathways implicating NF-κB. When activated, NF-κB traffics from the cytoplasm to the nucleus and binds to transcription sites, activating genes involved in the inflammatory response. Implicated genes include genes transcribing pro-inflammatory cytokines [tumor necrosis factor alpha [TNFa] and interleukin-1 [IL-1]), chemokines (intercellular adhesion molecule-1 [ICAM-1] and vascular cell adhesion molecule-1 [VCAM-1]), and pathways regulating the production of nitric oxide. The immune response is initially mediated by cells of the innate immune system, followed by the adaptive immune system. Both pro-inflammatory and anti-inflammatory responses are involved [12]. The balance between pro and anti-inflammatory mediators regulates the inflammatory process [13,14]. As inflammation progresses, additional organs may be affected, such as the cardiovascular, renal, or neurological systems. Coagulation abnormalities may also occur, leading to disseminated intravascular coagulation, thereby promoting additional tissue damage [15]. Respiratory involvement may also occur during sepsis, leading to acute respiratory distress syndrome, even when pneumonia is not the initial trigger of sepsis.

The endocrine response to stress involves stress hormones, notably glucocorticoids (cortisol), whose main purpose is to maintain homeostasis throughout the organism by optimizing the production of energy [16]. During stress, the hypothalamic-pituitary-adrenal axis increases cortisol production through increased production of CRH/ACTH [9]. Three key pathophysiologic features occur simultaneously during critical illness: dysregulation of the HPA axis (due to high level of pro-inflammatory cytokines [17]), altered cortisol synthesis at every stage (altered CRH/ACTH synthesis, altered adrenal steroidogenesis [18]) and altered peripheral metabolism of cortisol and tissue resistance to corticosteroids [19].

### 1.2. Acute Respiratory Distress Syndrome [ARDS]—Definition, Epidemiology and Pathophysiology

Acute respiratory distress syndrome, or ARDS, is defined by an acute onset of symptoms, bilateral radiographic opacities not fully explained by effusion, atelectasis or masses, and arterial hypoxemia by PaO_2_/FiO_2_ < 300 mm Hg (defined as mild if 200 < PaO_2_/FiO_2_ ≤ 300 on CPAP or PEEP ≥ 5 cm H_2_O; Moderate if 100 < PaO_2_/FiO_2_ ≤ 200 on PEEP ≥ 5 cm H_2_O; Severe if 100 ≤ PaO_2_/on PEEP ≥ 5 cm H_2_O], with no cardiogenic explanation) [20]. The incidence of ARDS in the US ranges from 64.2 to 78.9 cases/100,000 person-years [21]. Mortality associated with ARDS is estimated to be 27–37% [22]. A significant number of ARDS survivors experience functional limitations, such as muscle weakness, respiratory symptoms, anxiety, depression and cognitive impairment [23,24]. Furthermore, long term mortality remains significant in ARDS survivors, with mortality rates of 20–34% at 5 years [23].

Local injury such as caused by pneumonia is the most common cause of ARDS, however any systemic insult (Transfusion-related acute lung injury, severe traumatic injury, pancreatitis, etc.] may result in ARDS. Sepsis is one of the main risk factors of ARDS. In the LUNG-SAFE cohort, pneumonia was the cause of 59.4% of cases of ARDS while extrapulmonary sepsis was the cause of 16% of cases of ARDS [25]. SARS-CoV-2 pneumonia is another well documented cause of ARDS [25,26].

Lung involvement in early ARDS is characterized by diffuse alveolar damage, caused by pro-inflammatory mediators (such as TNF, IL1, IL6, IL8) [27] leading to the recruitment and activation of PMNs, which in turn release toxic mediators (reactive oxygen species and protease) [28]. This inflammatory process disrupts alveolar epithelium, alters basal membranes, impairs fluid resorption and promotes the accumulation in the alveolar space of proteins and blood cells [29], leading to surfactant inactivation, atelectasis, reduced aerated lung volume, intra pulmonary shunt and refractory hypoxemia [30]. ARDS subgroups were recently described. One ARDS subgroup exhibits increased inflammatory parameters, higher requirements of vasopressors as well as higher mortality and fewer days alive free from mechanical ventilation or organ failure free [31]. The administration of anti-inflammatory agents such as corticosteroids, particularly in this inflammatory subgroup of ARDS may be of particular interest.

### 1.3. Corticosteroids

Corticosteroids are anti-inflammatory agents implicated in several biological processes, including carbohydrate, protein and lipid metabolism, and hydro-electrolytic balance. Cortisol (also known as hydrocortisone), the main natural glucocorticoid in man is produced by the adrenal glands. Cortisol is converted into cortisone, mainly by the kidney. Corticosteroid metabolism is regulated by the hypothalamic-pituitary-adrenal axis, more specifically by the adrenocorticotropic hormone (ACTH), which in turn is under the control of the corticotrophin-releasing hormone (CRH). Synthetic corticosteroids have been in use for several decades, mainly as powerful anti-inflammatory agents. Knowledge regarding their biological effects derives from the work of Kendall, Reichstein and Hench [32]. Corticosteroids are distinguished by their glucocorticoid (anti-inflammatory) and mineralocorticoid effect [33]. Cellular effects of corticosteroids are mediated by both genomic and non-genomic signaling pathways [34]. The genomic pathway involves cortisol binding the glucocorticoid receptor (GR), translocation into the cytosol, where it regulates, both directly or indirectly the expression of target genes [35]. The main effect of glucocorticoids is to inhibit the expression of pro-inflammatory cytokines, notably through the sequestration of NF-κB [36]. Glucocorticoids also inhibit the production of chemokines (such as intercellular adhesion molecule-1 [ICAM-1] and vascular cell adhesion molecule-1 [VCAM-1]) [37].

Preclinical studies have shown that the expression of the glucocorticoid receptor is increased in LPS stimulated macrophages [38]. Deletion of the glucocorticoid receptor leads to greater mortality in animal models of sepsis [39,40]. The administration of corticosteroids decreases the level of pro-inflammatory cytokines (TNF-α, IL1 and IL6) [41] and improves survival, in rat and canine models of sepsis [42,43]. Human studies have produced similar results. In healthy volunteers challenged by LPS, the administration of corticosteroids reduces plasma levels of TNF-α [44], IL1 and IL6 [45]. In septic shock low dose corticosteroids led to improved outcomes in recent large scale trials [46,47,48,49]. In a multicenter placebo-controlled trial of 1241 patients with septic shock (APROCCHSS)*,* corticosteroids reduced the risk of 90-day mortality (43% vs. 49.1%, RR 0.88 (95% CI [0.78–0.99] *p* = 0.03)) [48]. In the ADRENAL placebo-controlled trial in 3658 patients with septic shock, corticosteroids did not reduce 90-day mortality (OR 0.95, 95% CI [0.82–1.10], *p* = 0.5) [50]. Main differences between these two trials included sicker patients at the time of enrollment in APROCCHSS compared to ADRENAL, different sources of sepsis (abdominal in ADRENAL vs. lung infections in APROCCHSS) and differences in the type of experimental treatment (continuous administration of hydrocortisone in ADRENAL vs. bolus administration of hydrocortisone in APROCCHSS; use of hydrocortisone in ADRENAL vs. combination of hydrocortisone and fludrocortisone in APROCCHSS). Subsequent meta-analysis of 61 trials showed that corticosteroids improved shock reversal, reduced 28-day mortality (RR 0.91 95% CI [0.84–0.99]) and hospital mortality (RR 0.90, 95% CI [0.82 to 0.99]) but not long term mortality (RR 0.97, 95% CI [0.91 to 1.03]) [51,52]. Accordingly, the latest surviving sepsis guidelines recommended using corticosteroids in adult septic shock with an ongoing requirement of vasopressors [53]. A recent multicenter placebo-controlled trial found that corticosteroids reduce 28-day mortality by −5.6 percentage points (95% CI, −9.6 to −1.7; *p* = 0.006) in 795 patients with severe community-acquired pneumonia [54].

## 2. Corticosteroids in ARDS

The efficacy of corticosteroids in ARDS has been a subject of controversy for decades [55,56]. In animal models of ARDS, corticosteroids decreased the expression of pro-inflammatory mediators in lung tissue, including TNF-a, IL-1a, IL-1b, IL-6 and IL-12 p40, and reduces lung injury through the reduction of oxygen radicals produced by neutrophils [57,58]. Beyond their anti-inflammatory effects during the acute phase of inflammation, corticosteroids also contributed to the resolution of inflammation, trough reprogramming effects on macrophages. Corticosteroids have been administered during two distinct phases of ARDS, during the early stage of ARDS when inflammation is expected to be most important and during late phase of ARDS, when lung fibrosis predominates. The biological and pathological characteristics of these two entities differ greatly, explaining the observed conflicting results in the effects of corticosteroids in these two distinct conditions [59].

### 2.1. Corticosteroids in Early Stage ARDS [Table 1]

The early phase of ARDS is characterized by major alveolar inflammation. Thus, corticosteroids, potent anti-inflammatory agents, are theoretically expected to be relevant treatment for ARDS. In practice, clinical trials found variably favorable, neutral or harmful effects of corticosteroids in ARDS.

In an ancillary analysis of a RCT focused on septic shock, Annane et al., found that 7-day treatment with low dose of steroids was significantly associated with better outcomes in septic shock associated with early septic ARDS in non-responders to short cosyntropin stimulation test [47]. In a trial of 24 ARDS patients, early corticosteroid treatment (methylprednisolone 2 mg/kg/d followed by progressive dose tapering over 32 days] was associated with a significant reduction in lung injury score (LIS) (*p* < 0.003 at 5 days) [60]. Similar findings were observed in a larger cohort (LIS 69.8% in placebo group vs. 35.7% in corticosteroids group; *p* = 0.02), with methylprednisolone 1 mg/kg/d (progressively tapered off over 28 days) [61]. In an Egyptian study, early administration of hydrocortisone in septic ARDS was associated with improved oxygenation parameters and LIS without achieving a survival benefit on day 28 [62]. Trial of short course of high dose corticosteroids (vs. placebo] found no evidence for improved 45-day mortality in adults with ARDS (60% vs. 63% *p* = 0.74) [63]. More recently, Villar et al., found that in ARDS, dexamethasone (20 mg IV daily between day 1 to 5, then 10 mg daily between day 6 to 10) compared to placebo, increased the number of ventilator-free days (between-group difference 4.8 days [95% CI 2.57 to 7.03]; *p* < 0.0001), and reduced mortality at day 60 (between-group difference −15.3% [−25.9 to −4.9]; *p* = 0.0047) [64].

### 2.2. Corticosteroids in Late-Stage ARDS

Late-stage ARDS is characterized histologically by ongoing inflammation with fibroproliferation, presence of hyaline membranes, and persistent diffuse alveolar damage, leading to prolonged mechanical ventilation and a higher risk of death [21]. Meduri et al., reported in 9 ARDS patients with pulmonary fibrosis, that high dose of methylprednisolone may improve the LIS [65]. Wajanaponsan et al., found that low dose methylprednisolone administered >14 days after onset of ARDS was associated with increased mortality rates at 60 and 180 days [66]. The largest multicenter placebo-controlled trial, found no evidence for difference in 60-day mortality with corticosteroids initiated for late-stage ARDS (36% vs. 27% *p* = 0.26) [67].

### 2.3. Dose and Type of Corticosteroid

Not all corticosteroids exhibit the same biological properties. The dose and type of corticosteroid may yield variable effects on patients’ outcomes. In a trial of 304 patients with sepsis, high doses of methylprednisolone led to numerically more patients with ARDS in corticosteroids vs. placebo (32% vs. 25% *p* = 0.1), fewer reversions of ARDS (31% vs. 61% *p* = 0.015), and a higher 14-day mortality (52% vs. 22% *p* = 0.04) [68]. In another ARDS trial, high doses of methylprednisolone [30 mg/kg every 6 h for 1 day] did not reduce mortality (*p* = 0.74) or reverse ARDS (*p* = 0.77) [63]. In another trial in patients with ARDS and critical illness related corticosteroids insufficiency, hydrocortisone administered 3 times a day (1 mg/kg/d) for seven days increased survival rates and reduced shock rate (5/12 vs. 10/14, *p* < 0.05), with no significant effect on 28-day mortality [69].

### 2.4. Pathogens

The use of corticosteroids in viral ARDS, especially in influenza related ARDS, remains controversial [70]. In a retrospective study, early course of corticosteroids may be associated with increased mortality risk (33.7% vs. 16.8%, *p* = 0.005) [71]. Worst outcomes with corticosteroids in influenza related ARDS may be related to the degree of viral replication, or to bacterial co-infections. Similarly, another observational study found increased ICU mortality with corticosteroids (27.5% vs. 18.8%, *p*  <  0.001) [72]. Observational studies in severe SARS and MERS infections, corticosteroids were not associated with improved outcomes [73,74,75,76]. These observational studies have high likelihood of selection bias with corticosteroids being given to the sickest patients [77]. Relying on indirect evidence from observational studies in flu, SRAS and MERS infections, initial recommendations were against administering corticosteroids in SARS-CoV-2 [78].

Patients suffering from COVID-19 exhibit elevated levels of inflammatory cytokines and chemokines, similarly to sepsis [79]. Corticosteroids were the first treatment to be effective in reducing mortality in oxygen dependent SARS-CoV-2 related acute respiratory failure [80]. Although studies of corticosteroids in COVID-19 acute respiratory failure did not use the Berlin definition of ARDS, their biological, clinical features and outcomes are highly similar [81]. RECOVERY was the first trial (*n* = 6245 patients enrolled) to demonstrate the survival benefits from dexamethasone in COVID-19. Among hospitalized patients, dexamethasone 6 mg for up to 10 days resulted in a lower 28-day mortality compared to usual care, especially in patients who were receiving mechanical ventilation (29.3% vs. 41.4%; RR 0.64; 95% CI [0.51 to 0.81]) [82]. These results were subsequently confirmed by several other trials [83,84,85,86] [Table 2], and a large meta-analysis of 7 trials accounting for 1703 patients (RR for 28-day mortality 0.79 95% CI [0.70 to 0.90]) [80].

Subsequent studies sought to determine whether higher doses of corticosteroids were more efficient in COVID-19. Munch et al., found that dexamethasone 12 mg/d did not improve the number of days alive without life support compared to standard doses (dexamethasone 6 mg/J) (adjusted mean difference, 1.3 days [95% CI, 0 to 2.6 days]; *p*  = 0.07) [87]. Bouadma et al., in COVIDICUS obtained similar results, higher doses of dexamethasone (20 mg/d for 10 d) did not significantly improve mortality compared to a standard dose (HR, 0.96 [95% CI, 0.69 to 1.33]; (*p* = 0.79)) [88]. Current guidelines recommend administrating dexamethasone 6 mg per day for 10 days in severe COVID-19 [89].

### 2.5. Adverse Events

The administration of corticosteroids may be associated with adverse events. In high-quality trials and meta-analyses in sepsis and in ARDS, indicate the main adverse events associated with corticosteroids may include neuromuscular weakness, gastrointestinal bleeding, hypernatremia and hyperglycemia [52,90]. A meta-analysis of 18 trials including 2826 ARDS patients, found no evidence for increased risk of muscular weakness: RR 0.85 95% CI [0.62 to 1.18] or gastrointestinal bleeding RR 1.20 95% CI [0.43 to 3.34], but increased risk of hyperglycemia RR 1.11 95% CI [1.01 to 1.23].

**Table 1 jcm-12-03340-t001:** Corticosteroids for early ARDS.

Author, Reference	Type	Sample Size	Study Population	Treatment	Results
Bernard et al. [63]	RCT, multicenter	99	ARDS as Partial pressure of oxygen ≤ 70 mm Hg on > 40% oxygen, PaO_2_/PAO_2_ ratio < 0.3, bilateral lung infiltrates, pulmonary artery wedge pressure ≤ 18 mm Hg	MPS 30 mg/kg IV 6 hourly for 24 h vs. placebo	PEP mortality MPS 30/50 (60%); Pl 31/49 (63.2) OR 0.75 [0.4 to 1.57] *p* = 0.74
Meduri et al. [60]	RCT multicenter	24	ARDS 1994 7 days of mechanical ventilation with an LIS of 2.5 or greater and less than a 1-point reduction from day 1 of ARDS, and no evidence of untreated infection.	MPS Loading dose of 2 mg/kg; then 2 mg/kg/d from day 1 to day 14, 1 mg/kg/d from day 15 to day 21, 0.5 mg/kg/d from day 22 to day 28, 0.25 mg/kg/d on days 29 and 30, 0.125 mg/kg/d on days 31 and 32. vs. placebo	PEP Lung injury and mortality day 10 MPS 1.7 [0.1]; Pl 3.0 [0.2]; *p* < 0.001 SEP: *Mortality* MPS 0/16 (0%); Pl 5/8 (62%) *p* = 0.002 *Mortality in hospital* MPS 2/16 (12.5%); Pl 5/8 (62.5%) OR 0.41 [0.06 to 99] *p* = 0.03
Steinberg et al., ARDSnetwork, [67]	RCT Multicenter	132/180	ARDS 1994 in early and late stage At least 7 days duration ARDS; *p*/F < 200 Intubated, mechanical ventilation	MPS Loading dose of 2 mg/kg of predicted body weight followed by 0.5 mg/kg 6 hourly for 14 days; 0.5 mg/kg 12 hourly for 7 days; and then tapering of the dose.	In early ARDS (7–13 d) PEP mortality at 60 days MPS (36%); Pl (27%) *p* = 0.26
Annane et al. [47]	post Hoc RCT	129/300 177 ARDS: 129 non responders, 48 responder	ARDS 1994 bilateral infiltrate on chest radiography, PaO_2_/FiO_2_ < 200 mm Hg and Pulmonary occlusion pressure ≤ 18 mm Hg or no clinical evidence of left atrial hypertension	HSHC 50 mg IV 6 hourly and 9-alpha fludrocortisone once a day for 7 days.	PEP: mortality at 28-day In the non-responder subgroup HSHC + FC 33/62 (53%); Pl 50/67 (75%) RR = 0.71; 95% CI [0.54 to 0.94] *p* = 0.013 OR = 0.35; 95% CI [0.15 to 0.82], *p* = 0.016). In the responder group No significant result HSHC + FC 16/23 (70%); PL 12/25 (48%) RR = 1.4; 95% CI [0.89 to 2.36] *p* = 0.130 OR = 2.29; 95% CI [0.49 to 10.64] *p* = 0.290
Meduri et al. [61]	RCT multicenter	91	ARDS 1994 Intubated and Mechanical ventilation ARDS ≤ 72 H of study entry	MPS Loading dose of 1 mg/kg Then 1 mg/kg/d from day 1 to day 14, 0.5 mg/kg/d from day 15 to day 21, 0.25 mg/kg/d from day 22 to day 25, 0.125 mg/kg/d from day 26 to day 28.	PEP 1-point reduction in LIS or MPS 69.8% vs. Pl 35.7%; *p* = 0.002 successful extubation 7-day MPS 53.9% vs. Pl 25.0%; *p* = 0.01
Tongyoo et al. [91]	RCT Single center	197	Severe sepsis or septic shock receiving IMV for hypoxemic respiratory failure within 12 H of study entry + ARDS 1994 then reclassified accordingly to ARDS 2012	HSHC 50 mg every 6 h or placebo	PEP 28 day all-cause mortality HSHC (22.5%) vs. Pl (27.3%) RR 0.82 [0.50 to 1.34] *p* = 0.51 HR 0.80, 95% CI [0.46 to 1.41]; *p* = 0.44
Villar et al., DEXA-ARDS, [64]	RCT multicenter	277/314 stopped low enrollment 88%	ARDS 2012 (but PEEP ≥ 10) Moderate to severe ARDS < 24 h (but PEEP ≥ 10)	DXM IV 20 mg once daily day 1 to 5 then 10 mg once daily day 6 to 10	PEP N° ventilator-free from day of randomization to day 28 Between-group difference 4.8 days 95% CI [2.57 to 7.03]; *p* < 0.0001).
Horby et al., RECOVERY, [82]	RCT multicenter	6425	Hospitalized patients with suspected or laboratory confirmed COVID-19	DXM 6 mg (IV or orally) during 10 days vs. usual care	PEP 28 d mortality Overall: DXM 482/2104 (22.9%); Pl 1110/4321 (25.7%) (age-adjusted RR 0.83; 95% CI [0.75 to 0.93]; *p* < 0.001) >sub group mechanical ventilation (1007): 29.3% vs. 41.4%; RR 0.64; 95% CI [0.51 to 0.81]
Tomazini et al., CoDEX, [84]	RCT multicenter	299/350	COVID-19 infection suspected or confirmed, receiving IMV within 48H of meeting criteria for moderate to severe ARDS 2012	DXM 20 mg daily for 5 days followed by 10 mg daily for 5 days	PEP Ventilator-free days (alive + free from IMV) DXM 6.6 95% CI [5.0 to 8.2) vs. Pl 4.0 95% CI [2.9 to 5.4] difference 4.0 95% CI [2.9 to 5.4]
Dequin et al., CAPE COVID [92]	RCT multicenter	149/290	Confirmed or suspected SARS-CoV-2 + 1 severity criteria IMV (PEEP > 5 cm H_2_O), *p*/F < 300 HFOT > 50% Fi, PaO_2_/FiO_2_ < 300 FMOT (specified charts), PSI > 130	HSHC 200 mg daily for 4 to 7 then 100 mg daily for 2 to 4 days then 50 mg daily for 2 to 3 days total 8 days	PEP: 21-day treatment failure (death or persistent dependency on mechanical ventilation or high-flow oxygen therapy HSHC 42.1% vs. pl 50.7% Difference −8.6% [95.48% CI, −24.9% to 7.7%]; *p* = 0.29)
Angus et al., REMAP CAP- [85]	RCT multicenter	384	COVID-19 suspected or confirmed, severe ICU for Respiratory failure (invasive or non-invasive IMV or HFN flow rate > 30 L/m, and FI > 40% Cardiovascular failure: vaopressor/inotrope	3 randomization arms Fixed: HSHC 50 mg every 6 h daily for 7 days Shock: HSHC 50 mg/6 h for 7 days while in shock No HSHC Or 200 mg/6 h for 7 days	PEP Composite of hospital mortality and ICU organ support-free days to day 21 Fixed 0 QR, −1 to 15; OR 1.43 95% CI [0.91 to 2.27] Shock 0 IQR, –1 to 13; OR1.22 95% CI [0.76–1.94] None 0 0 (IQR, −1 to 11)
Barros et al., MetCOVID [86]	RCT single center	246	Clinical-radiological suspicion of COVID-19 Sat ≤ 94% in room air or Requiring O_2_ or IMV	MPS IV 0.5 mg/kg every 12 h × 5 days	PEP pulmonary function testing at day 120 follow-up visit. (Pulmonary function and maximal respiratory pressure testing, DASI, 6MWT) FEV1 (2.6, [0.7], *p* = 0.01) and FVC (3.2, [0.8], *p* = 0.01
Dequin et al., CAPE COD, [54]	RCT multicenter	795	Severe community-acquired pneumoniae, defined by the presence of at least one of four following criteria The initiation of MV (invasive or noninvasive) with a positive end-expiratory pressure level ≥ 5 cm of water The initiation of the administration of oxygen through a HFOT with a ratio of PaO_2_:FiO_2_ < 300, with a FiO_2_ of 50% or more; For patients wearing a non-rebreathing mask, an estimated PaO_2_:FiO_2_ ratio < 300, or a score of more than 130 on the Pulmonary Severity Index, which classifies patients with community-acquired pneumonia into five groups according to increasing severity, with a score of more than 130 defining group V	HSHC continuous IV 200 mg/day during the first 4 days. On day 4, regarding medical decision based on predefined criteria, following administration for a total of 8 or 14 days	PEP mortality at day 28 HSHC 25 of 400 patients 6.2%; 95% CI, [3.9 to 8.6] vs. placebo 47 of 395 patients 11.9%; 95% CI, [8.7 to 15.1] (Absolute difference, −5.6 percentage points; 95% CI, [−9.6 to −1.7]; *p* = 0.006).

PEP: primary end point; SEP: secondary end point; MPS: Methylprednisolone; HSHC: hemisuccinate hydrocortisone; DXM: dexamethasone ARDS: acute respiratory distress syndrome; AHRF: acute hypoxemic respiratory failure, IMV: invasive mechanical ventilation; LIS: lung injury score; HFOT = High flow oxygen therapy; FMOT: face max oxygen therapy; PaO_2_/FiO_2_: partial pressure of oxygen/fraction of inspired oxygen; ARDS AEEC 1994: acute syndrome, hypoxemia PaO_2_/FiO_2_ < 200, bilateral infiltrate on chest radiography, cannot be due to cardiac failure [elevated left atrial pressure] assessed by clinical examination or a PCWP > 18 cm H_2_O; ARDS Berlin 2012: ≤7 days since onset of predisposing clinical definition, bilateral opacities on X-ray or CT scan no attributed to pleural effusion, atelectasis or nodules, respiratory failure that cannot be attributed to heart failure or volume overload, PaO_2_/FiO_2_ with use of ≥5 cm H_2_O of PEEP ([201–300 mm Hg] as mild ARDS; [101–200 mm Hg] as moderate ARDS; <100 mm Hg as severe ARDS) and perform additional studies to rule out cardiogenic oedema (echocardiography, BNP).

**Table 2 jcm-12-03340-t002:** Dose effect in septic ARDS and SARS-CoV-2 Acute respiratory failure—RCT.

Dose	Author, Reference	Type	Sample Size	Study Population	Treatment	Results
High Dose	Bone et al. [68]	RCT multicenter	382 (304 randomized)	Sepsis define as fever or hypothermia rectal T° >38.3 or <35.5 °C tachypnea (>20 bpm), tachycardia (>90 bpm) one of the following indices of organ dysfunction: a change in mental status, hypoxemia, elevated lactate levels or oliguria	MPS, 30 mg/kg, every six hours for 48 h vs. placebo	PEP: ARDS development MPS 50/152 32% Pl: 38/152 25% (*p* = 0.1) SEP: Reverse ARDS MPS 15/50 (31%); Pl: 23/38 (61%) (*p* = 0.005) 14 d mortality MPS 26/50 (52%); Pl 8/22 (%) (*p* = 0.004)
	Bernard et al. [63]	RCT multicenter	99	ARDS Partial pressure of oxygen ≤70 mm Hg on >40% oxygen, PaO_2_/PAO_2_ ratio <0.3, bilateral lung infiltrates, pulmonary artery wedge pressure ≤18 mm Hg	MPS 30 mg/kg IV 6 hourly for 24 h vs. placebo	PEP mortality MPS 30/50 (60%); Pl 31/49 (63.2) OR 0.75 [0.4 to 1.57] *p* = 0.74 SEP reversal of ARDS MPS 18/50 (39%); Pl 19/49 (36%) (*p* = 0.07)
High- Moderate Dose	Tomazini et al., CoDEX, [84]	RCT multicenter	299/350	COVID-19 moderate to severe ARDS according to Berlin, mechanical ventilation	DXM 20 mg daily for 5 days followed by 10 mg daily for 5 days	PEP Ventilator-free days (alive + free from IMV) DXM 6.6 95% CI% [5.0 to 8.2] vs. Pl 4.0 95% CI [2.9 to 5.4] difference 4.0 95% CI [2.9 to 5.4] SEP 28 d mortality all cause no difference
	Barros et al., MetCOVID, [86]	RCT single center	246	In hospital COVID-19 Requiring O_2_ or IMV	MPS IV 0.5 mg/kg every 12 h × 5 days	PEP pulmonary function testing at day 120 follow-up visit. (pulmonary function and maximal respiratory pressure testing, DASI, 6MWT) FEV1 (2.6, [0.7], *p* = 0.01) and FVC (3.2, [0.8], *p* = 0.01
	Munch et al., COVIDSTEROID, [87]	RCT multicenter	982/1000	Confirmed COVID-19 with > 10 L/mn O_2_ or IMV	DXM 12 mg/d vs. DXM 6 mg/d for 10 days	PEP number of days alive without life support at 28 d 22 d vs. 20.5 d adjusted mean difference, 1.3 days 95% CI [0 to 2.6 days]; *p* = 0.07
	Bouadma et al., COVIDICUS, [88]	RCT multicenter	546	Admitted to ICU within 48H for confirmed or highly suspected COVID-19 AHRF (PaO_2_ < 70 mm Hg, SpO_2_ < 90%, tachypnea > 30 mn, labored breathing, respiratory distress, or need of O_2_ flow ≥ 6 L/mn)	DXM 6 mg/d for 10 days (or placebo prior to RECOVERY result communication) or Dexamethasone 20 mg/d for 5 days followed by Dexamethasone 10 mg/d for 5 days	PEP day 60 all-cause mortality Low 26.8% vs. High 25.9% Absolute risk difference −0.8% 95% CI [−8.3 to 6.5] HR, 0.96 95% CI [0.69 to 1.33]; (*p* = 0.79)
Low Dose	Dequin et al., CAPE COVID, [92]	RCT multicenter	149/290	Confirmed or suspected SARS-CoV-2 + 1 severity criteria IMV (PEEP > 5 cm H_2_O), *p*/F < 300 HFOT > 50% Fi, PaO_2_/FiO_2_ < 300 FMOT (specified charts), PSI > 130 Excluded septic shock Low recruitment	HSHC 200 mg daily for 4 to 7 then 100 mg daily for 2 to 4 days then 50 mg daily for 2 to 3 days total 8 days	PEP: 21-day treatment failure (death or persistent dependency on mechanical ventilation or high-flow oxygen therapy HSHC 42.1% vs. pl 50.7% Difference −8.6% 95.48% CI [−24.9% to 7.7%]; *p* = 0.29) SEP 21 d mortality HSHC 14.7% vs. placebo 27.4% difference −12.7% 95% CI [−25.7% to 0.3%]; *p* = 0.06
	Angus et al., REMAP CAP, [85]	RCT multicenter	384	COVID-19 suspected or confirmed, severe ICU for Respiratory failure (invasive) or noninvasive MV or HFN flow rate > 30 L/m, and FI > 40% Cardiovascular failure: vasopressor/inotrope Stop ethical	3 randomization arms Fixed: HSHC 50 mg every 6 h daily for 7 days Shock: HSHC 50 mg/6 h for 7 days while in shock No HSHC Or 200 mg/6 h for 7 days	PEP Composite of hospital mortality and ICU organ support-free days to day 21 Fixed 0 QR, −1 to 15; OR 1.43 95% CI [0.91 to 2.27] Shock 0 IQR, −1 to 13; OR1.22 95% CI [0.76 to 1.94] None 0 0 (IQR, −1 to 11) SEP 28 d mortality Fixed 30%; OR 1.03 95% CI [0.53 to 1.95] Shock 26%;1.10 95% CI [0.58 to 2.11] None 33%
	Horby et al., RECOVERY, [82]	RCT multicenter	6425	hospitalized patients with suspected or laboratory confirmed COVID-19	DXM 6 mg (IV or orally) for 10 days vs. usual care	PEP 28 d mortality Overall: DXM 482/2104 (22.9%); Pl 1110/4321 (25.7%) (age-adjusted RR 0.83; 95% CI [0.75 to 0.93]; *p* < 0.001) >sub group mechanical ventilation (1007): 29.3% vs. 41.4%; RR 0.64; 95% CI [0.51 to 0.81) SEP Invasive mechanical ventilation or death RR 0.93; 95% CI [0.85 to 1.01]
	Corral- Gudino et al., GLUCOCOVID, [93]	RCT multicenter	64/180 Low recruitment	Symptom > 7 days Radiological evidence of lung disease on chest X-ray or CT-scan Moderate to severe disease with abnormal gas exchange: (PaO_2_/FiO_2_ or PaFi) < 300 Or (SaO_2_/FiO_2_ or SaFi) < 400 Or ≥2 criteria of the BRESCIA-COVID Respiratory Severity Scale (BCRSS Evidence of a systemic inflammatory response: serum C-reactive protein > 15 mg/dL, D-dimer > 800 ng/mL, ferritin > 1000 mg/dL, or IL-6 levels > 20 pg/mL	MPS 40 mg IV q12 × 3 days and then 20 mg q12 h × 3 days	PEP: Composite endpoint (in-hospital all-cause mortality, escalation to ICU admission, or progression of respiratory insufficiency that required noninvasive ventilation) In ITT: MPS 40% vs. Pl 48% *p* = 0.25 In PP: RR 0.42 95% CI [0.20 to 0.89]

PEP: primary end point; SEP: secondary end point; MPS: Methylprednisolone; HSHC: hemisuccinate hydrocortisone; DXM: dexamethasone ARDS: acute respiratory distress syndrome; AHRF: acute hypoxemic respiratory failure, IMV: invasive mechanical ventilation; LIS: lung injury score; HFOT = High flow oxygen therapy; FMOT: face max oxygen therapy; PaO_2_/FiO_2_: partial pressure of oxygen/fraction of inspired oxygen; ARDS AEEC 1994: acute syndrome, hypoxemia PaO_2_/FiO_2_ < 200, bilateral infiltrate on chest radiography, cannot be due to cardiac failure (elevated left atrial pressure) assessed by clinical examination or a PCWP > 18 cm H_2_O; ARDS Berlin 2012: ≤7 days since onset of predisposing clinical definition, bilateral opacities on X-ray or CT scan no attributed to pleural effusion, atelectasis or nodules, respiratory failure that cannot be attributed to heart failure or volume overload, PaO_2_/FiO_2_ with use of ≥5 cm H_2_O of PEEP ([201–300 mm Hg] as mild ARDS; [101–200 mm Hg] as moderate ARDS; <100 mm Hg as severe ARDS) and perform additional studies to rule out cardiogenic oedema (echocardiography, BNP).

## 3. Summary

Multiple studies have yielded variable effects of corticosteroids in ARDS. Discrepancies between studies may be explained by multiple factors such as differences in study design, causes of ARDS, dose, timing of initiation, duration, modalities of treatment cessation, and type of corticosteroids.

Despite decades of controversies about the use of corticosteroids in ARDS, the demonstration that corticosteroids saved numerous lives during the COVID pandemic renewed the interest in investigating the benefit-to-risk balance of corticosteroids in ARDS. Likewise, the recent characterization of ARDS phenotypes and endotypes delineates treatable traits possibly relevant to corticotherapy [94]. Alveolar procollagen may be an indicator of fibroproliferation, which may help identify non-resolving ARDS that is likely to respond to treatment by corticosteroids [95]. The analysis of expired air in ARDS is also a promising venue of research [96].

## 4. Conclusions

Corticosteroids are effective treatment for severe community-acquired pneumonia, septic shock and COVID-19-related acute respiratory failure. High-quality trials are awaited before a broader use of corticosteroids in ARDS.

## Data Availability

No new data were created or analyzed in this study. Data sharing is not applicable to this article.

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
