# Peer review of "Corticosteroids in ARDS"

_jcm, 2023, doi:10.3390/jcm12093340_

Round 1

Reviewer 1 Report

Thank you for referring to me the manuscript “Corticosteroids in septic ARDS”. 

In this review paper, the authors comprehensively discuss the role of corticosteroid treatment in critical clinical scenarios like sepsis and ARDS. 

The article is well-written and has a well-organized structure, starting with epidemiology, definitions, and pathophysiology. In the second part, the results of the use of corticosteroids in clinical conditions are described with all pros and cons regarding its use, which makes the review objective.  

The text is enhanced with the tables with all important trials, which may be very helpful in exploring this topic.

In summary, the authors did an important work preparing this interesting revision on the controversial subject, which is the use of corticosteroids in sepsis and ARDS. This review may help clinicians make clinical decisions, and researchers plan future trials. 

I recommend publishing this article without any additional improvement apart from some typos, e.g. SARS should be written with capital letters.

Author Response

Thank you for your comments.

As suggested we have extensively revised the manuscript for English language and grammar.

Reviewer 2 Report

Dear editor,

Thank you for the opportunity to review the manuscript. The paper touches an important filed where many recent studies have provided new insights during the recent COVID pandemic.

I would however recommend the following changes to improve the message of the review:

1)      The section on effect of corticosteroids in sepsis seems unmotivated and does not per se provide enough insight nor cover the topic with enough substance. Would recommend to either skip this section or expanded to cover more broadly the effects of steroids in conditions with substantial risk of ARDS.

2) The review omits the recently published CAPE-COD trial?

3)      The discussion is unfocused and is merely a repetition of the previous sections. Would recommend to thoroughly discuss: 1) The causes for discrepancy between trials, 2) The insights gained from COVID, 3) Potential for personalized treatment and how biomarkers / phenotypes could improve the administration of steroids, 4) Discus future study questions and design in much deeper details.

4)      Would recommend language review by native speaking.

Author Response

Thank you for the opportunity to review the manuscript. The paper touches an important filed where many recent studies have provided new insights during the recent COVID pandemic.

1)     The section on effect of corticosteroids in sepsis seems unmotivated and does not per se provide enough insight nor cover the topic with enough substance. Would recommend to either skip this section or expanded to cover more broadly the effects of steroids in conditions with substantial risk of ARDS.

As suggested, we have comprehensively rewritten the chapter on corticosteroids in sepsis, notably including a paragraph on the effect of corticosteroids in severe community acquired pneumonia. We have also combined this paragraph with the previous one on the cellular effects of corticosteroids.

2) The review omits the recently published CAPE-COD trial?

We agree with the reviewer’s assessment. Accordingly, we have added the Dequin et al trial page 4 line 165-169

3)      The discussion is unfocused and is merely a repetition of the previous sections. Would recommend to thoroughly discuss: 1) The causes for discrepancy between trials, 2) The insights gained from COVID, 3) Potential for personalized treatment and how biomarkers / phenotypes could improve the administration of steroids, 4) Discus future study questions and design in much deeper details.

As suggested the discussion has been completely overhauled. As requested, we have elaborated the 4 points mentioned by the reviewer:

“Multiple trials and observational studies have been conducted assessing the effect of corticosteroids in ARDS with conflicting results. Discrepancy between studies may be explained by multiple factors. First by methodological difference, influenza, MERS and SARS studies being retrospective, these studies are prone to bias. Other factors may also be in play such as a different pathobiology compared to SARS-CoV-2, with greater viral replication and less inflammation. Second differences may be due to the heterogeneity in the target population. Corticosteroids administered during the acute phase of ARDS aim at countering the excessive inflammatory process, while corticosteroids administered in prolonged ARDS aim at countering the fibroproliferative state, which have distinct biological causes. Variability between trials may also be related to the type/dose of corticosteroid being administered. Dexamethasone for instance does not exhibit any mineralocorticoid effect while hydrocortisone and methylprednisolone do. Trails assessing the efficacy of methylprednisolone tend to administer high doses of corticosteroids followed by prolonged periods of tapering which may increase the risk of immunosuppression and severe side effect.

Although the use of corticosteroids in ARDS has been a matter of debate for decades, the COVID-19 pandemic has significantly altered the debate. During the pandemic, high quality randomized controlled trials demonstrated the benefit of corticosteroids in severe forms of COVID-19. The pandemic also helped ascertain the importance of high quality randomized controlled compared to observational, uncontrolled studies. Awake prone positioning was successfully administered during the COVID-19 pandemic and may have a role to play in non COVID acute respiratory failure/ARDS. Novel trial designs, most notably platform, adaptive trials (RECOVERY, REMAP-CAP) are flexible tools that were highly successful during the COVID-19 pandemic and are being built upon.

Personalized medicine has previously been implemented in the field of oncology with great success. However, personalized medicine is not yet routinely used in the ICU. Past trials assessing the use of corticosteroids in ARDS included all patients using a one size fits all approach. Since ARDS is a syndrome and not a disease; the etiology, biology, physiology, imaging characteristics and prognosis are heterogenic. The identification of specific ARDS phenotypes or biomarkers may help identify treatable traits. Treatable traits being investigated in ARDS include the inflammatory phenotype. A hyperinflammatory phenotype is the most likely to respond to anti-inflammatory agents such as corticosteroids, whereas an immunosuppressed phenotype would cause anti-inflammatory agents to be withheld. Several biomarkers or phenotypes, including artificial intelligence derived algorithms predicting a favorable response to corticosteroids in sepsis may be assessed in ARDS. Alveolar procollagen may be an indicator of fibroproliferation, which may help identify non-resolving ARDS, susceptible to respond to treatment by corticosteroids. The analysis of expired air in ARDS is also a promising venue of research.

Multiple future study questions have arisen since the COVID-19 pandemic. The first such question is to determine the role of immunomodulatory agents (corticosteroids, anti IL6 receptors, tyrosine kinase inhibitors) in non COVID-19 ARDS, which have proven to be efficient and safe in severe forms of SARS-CoV-2 acute respiratory failure. Future trials should aim at preventing the appearance of the condition by identifying and treating patients at risk of developing ARDS or in the early stages of acute respiratory failure. Platform trials are being developed to investigate ARDS as well as other allied conditions”

4)      Would recommend language review by native speaking.

As suggested, one of the authors (NH) a native English speaker has extensively revised the manuscript for grammar and English language.

Reviewer 3 Report

Thank you very much for the opportunity to review this very informative article conveying the current state of the art when it comes to steroids for "septic" ARDS, though the authors--including Dr. Annane who is one of the world's authorities--seem to venture out to venture into the topic of steroids in ARDS of all types.  In reality, I see this review as having been divided into 3 reasonable sections: steroids in septic shock, steroids in ARDS, steroids in viral pneumonia.  I think this is a very legitimate way to handle the topic, but I wonder whether the article's title accurately conveys the actual contents of the article.  With that as the starting point, I offer the following points for the authors' consideration:

1. In the Introduction, the authors seem to switch back and forth between epidemiological data from the US and global data.  If referring to global epidemiology, I would be explicit about that.  For example, I assume that diarrheal disease was the commonest cause of sepsis in 2017 globally and not specifically in the US.

2. Please avoid one-sentence paragraphs such as lines 43-44.

3. Line 96: would be more specific than just psychological "issues."

4. Related to #3 above, I think reviews should minimize citations of other reviews whenever possible.  For example, there are robust RCTs that can be cited in place of reference #23 (e.g., PubMed ID 21470008).

5. Line 97: long term mortality is higher in ARDS survivors--higher than in whom?

6. Line 100: authors did not close parentheses.

7. Line 180: authors may want to update that reference with the more fitting recently published CAPE COD trial in NEJM

8. When referring to studies by author name (e.g., Annane), I would add et al in each case to be clear that there was a group of authors.

9. Reference 70 seems to be an article in Chinese.  If that's the case, the authors should specify that in the reference list.

10. Lines 294-5: "Even definition was also not consensual in those various trials..." The meaning of this statement is not clear.

11. I am not quite sure what is the role of a Discussion section in a review article: the entire article is in essence one long Discussion.  Perhaps Summary would be a better title for that section.

12. Line 341: Would replace "First" with "Earlier."

13. Finally, to circle back to my initial comments at the top, I found it interesting that the authors chose to cover viral PNA/ARDS in a review article on "septic" ARDS because viral infection is not what I would typically associate with the term "septic" ARDS, whatever that means. To me, "septic" ARDS would be more reflective of ARDS in the setting of bacterial PNA or extrapulmonary bacterial infection. Often, it would be ARDS in association with septic shock. I don't think this otherwise fairly long review article would suffer much from the exclusion of the discussion of viral ARDS, which probably deserves a separate review article, but I recognize that the article and its table are already written to include this topic, so perhaps there are other ways to address this. One might be to entitle the article "Corticosteroids in ARDS" and leave out the word "septic." Secondly, if keeping the viral discussion, then what should be added is an attempt to reconcile the seemingly conflicting data for steroids in influenza ARDS compared to SARS CoV-2 and even arguably to SARS. I don’t think this distinction can just be glossed over because the adverse influenza ARDS data for steroids created a lot of controversy and opposition to steroids early in the pandemic, and the harm seems to be real in that condition. Part of the discussion should include potential reasons for this dichotomy: is it the propensity for early superinfection in flu ARDS that doesn't exist in SARS//COVID, is it a matter of pathobiology, tempo of evolution of underlying lung histology, etc? I recognize that the distinction may not be fully understood, but I think it should at least be acknowledged. We attempted to try to understand the seemingly differential response to steroids in flu ARDS compared to COVID in the early months of the pandemic--at that time of course most of the arguments were purely speculative (PMID 34542192). 

Author Response

Thank you very much for the opportunity to review this very informative article conveying the current state of the art when it comes to steroids for "septic" ARDS, though the authors--including Dr. Annane who is one of the world's authorities--seem to venture out to venture into the topic of steroids in ARDS of all types.  In reality, I see this review as having been divided into 3 reasonable sections: steroids in septic shock, steroids in ARDS, steroids in viral pneumonia.  I think this is a very legitimate way to handle the topic, but I wonder whether the article's title accurately conveys the actual contents of the article.  With that as the starting point, I offer the following points for the authors' consideration:

  1. In the Introduction, the authors seem to switch back and forth between epidemiological data from the US and global data. If referring to global epidemiology, I would be explicit about that. For example, I assume that diarrheal disease was the commonest cause of sepsis in 2017 globally and not specifically in the US.

Most of epidemiological data in this section are global except when specifically marked, notably concerning the cost of sepsis. As suggested by the reviewer, we have added “globally” whenever relevant (page 1 line 36 to 42)

  1. Please avoid one-sentence paragraphs such as lines 43-44.

Thank you for pointing this out. Modifications have been made to avoid one-sentence paragraphs throughout the manuscript.

  1. Line 96: would be more specific than just psychological "issues."

As suggested, we replaced “psychological issues” by anxiety and depression (page 2 line 96)

And 4. Related to #3 above, I think reviews should minimize citations of other reviews whenever possible.  For example, there are robust RCTs that can be cited in place of reference #23 (e.g., PubMed ID 21470008).

As suggested, we changed reference #23 by Herridge MS, Tansey CM, Matté A, Tomlinson G, Diaz-Granados N, Cooper A, et al. Functional disability 5 years after acute respiratory distress syndrome. N Engl J Med. 2011 Apr 7;364(14):1293–304 (page 2 line 96)

  1. Line 97: long term mortality is higher in ARDS survivors--higher than in whom?

Thank you for pointing this out, we have changed the sentence to “long term mortality remains significant in ARDS survivors, with mortality rates of 20-34% at 5 years” (page 2 line 96-97)

  1. Line 100: authors did not close parentheses.

Thank you for pointing this out, the parentheses have been closed (page 3 line 101)

  1. Line 180: authors may want to update that reference with the more fitting recently published CAPE COD trial in NEJM

We agree with the reviewer’s assessment. Accordingly, we have added a reference to theDequin et al trial in page 4 line 165-169

  1. When referring to studies by author name (e.g., Annane), I would add et al in each case to be clear that there was a group of authors.

Thank you for pointing this out, as suggested we have updated the study name (page 4 line190)

  1. Reference 70 seems to be an article in Chinese. If that's the case, the authors should specify that in the reference list.

The reviewer is corrected, Liu et al trial is in Chinese; we have specified this point in the reference list.

  1. Lines 294-5: "Even definition was also not consensual in those various trials..." The meaning of this statement is not clear.

The reviewer is correct, and we have revised body text as follows page 6 on line 282-286:  « In a meta-analysis of 18 RCT including 2826 ARDS patients, muscular weakness: RR 0.85 95%CI [0.62 to 1.18], gastrointestinal bleeding RR 1.20 95%CI [0.43 to 3.34] were not found to be associated with corticosteroid administration. Only hyperglycemia RR 1.11 95%CI [1.01 to 1.23] was found to be associated with the administration of corticosteroids. »

  1. I am not quite sure what is the role of a Discussion section in a review article: the entire article is in essence one long Discussion. Perhaps Summary would be a better title for that section.

We agree with the reviewer’s assessment. As suggested we change the section heading to Summary. (Page 6 line 315)

  1. Line 341: Would replace "First" with "Earlier."

As suggested  in 13., We have modified significantly the “discussion” Page 6 line 315 to 364

  1. Finally, to circle back to my initial comments at the top, I found it interesting that the authors chose to cover viral PNA/ARDS in a review article on "septic" ARDS because viral infection is not what I would typically associate with the term "septic" ARDS, whatever that means. To me, "septic" ARDS would be more reflective of ARDS in the setting of bacterial PNA or extrapulmonary bacterial infection. Often, it would be ARDS in association with septic shock. I don't think this otherwise fairly long review article would suffer much from the exclusion of the discussion of viral ARDS, which probably deserves a separate review article, but I recognize that the article and its table are already written to include this topic, so perhaps there are other ways to address this. One might be to entitle the article "Corticosteroids in ARDS" and leave out the word "septic." Secondly, if keeping the viral discussion, then what should be added is an attempt to reconcile the seemingly conflicting data for steroids in influenza ARDS compared to SARS CoV-2 and even arguably to SARS. I don’t think this distinction can just be glossed over because the adverse influenza ARDS data for steroids created a lot of controversy and opposition to steroids early in the pandemic, and the harm seems to be real in that condition. Part of the discussion should include potential reasons for this dichotomy: is it the propensity for early superinfection in flu ARDS that doesn't exist in SARS//COVID, is it a matter of pathobiology, tempo of evolution of underlying lung histology, etc? I recognize that the distinction may not be fully understood, but I think it should at least be acknowledged. We attempted to try to understand the seemingly differential response to steroids in flu ARDS compared to COVID in the early months of the pandemic--at that time of course most of the arguments were purely speculative (PMID 34542192).

As suggested by the reviewer, we have changed the title to Corticosteroids in ARDS.  We have also significantly expanded the discussion regarding the difference between influenza and SARS-CoV2 ARDS line 317 to 331 page 6. We have also added ref PMID 34542192 to the manuscript (page 6 line 250 to 253)

Round 2

Reviewer 2 Report

The paper has been substantially improved. No further comments.

Author Response

no additional changes were requested